# Machine Learning for Human Motion Intention Detection

**DOI:** 10.3390/s23167203

**Published:** 2023-08-16

**Authors:** Jun-Ji Lin, Che-Kang Hsu, Wei-Li Hsu, Tsu-Chin Tsao, Fu-Cheng Wang, Jia-Yush Yen

**Affiliations:** 1Department of Mechanical Engineering, National Taiwan University, No. 1, Sec. 4, Roosevelt Rd., Taipei City 106319, Taiwan; 2School and Graduate Institute of Physical Therapy, National Taiwan University, No. 17, Xuzhou Rd., Zhongzheng Dist., Taipei City 100025, Taiwan; 3Mechanical and Aerospace Engineering, Samueli School of Engineering, University of California, Los Angeles, CA 90095, USA; 4Department of Mechanical Engineering, National Taiwan University of Science and Technology, No. 43, Keelung Rd., Sec. 4, Da’an Dist., Taipei City 106335, Taiwan

**Keywords:** human intention detection, human–robot interaction, feedforward neural network (FNN), long short-term memory (LSTM), inertial measurement unit (IMU)

## Abstract

The gait pattern of exoskeleton control conflicting with the human operator’s (the pilot) intention may cause awkward maneuvering or even injury. Therefore, it has been the focus of many studies to help decide the proper gait operation. However, the timing for the recognization plays a crucial role in the operation. The delayed detection of the pilot’s intent can be equally undesirable to the exoskeleton operation. Instead of recognizing the motion, this study examines the possibility of identifying the transition between gaits to achieve in-time detection. This study used the data from IMU sensors for future mobile applications. Furthermore, we tested using two machine learning networks: a linearfFeedforward neural network and a long short-term memory network. The gait data are from five subjects for training and testing. The study results show that: 1. The network can successfully separate the transition period from the motion periods. 2. The detection of gait change from walking to sitting can be as fast as 0.17 s, which is adequate for future control applications. However, detecting the transition from standing to walking can take as long as 1.2 s. 3. This study also find that the network trained for one person can also detect movement changes for different persons without deteriorating the performance.

## 1. Introduction

The collection and analysis of body signals for human intent detection (HID) are critical for collaborative robotic applications [1,2], and the speedy and accurate prediction of human intent is essential to the success of the application [3]. Many research efforts have addressed the issue of accurate identification. This research, on the other hand, tries to address the issue of how fast one can detect a change in movement.

This paper addresses the detection of motion changes in the lower limbs. The ability to detect lower-limb motion change can help direct the exoskeleton controller to determine the proper gaits that would not conflict with the pilot’s intention. A conflicting maneuver, especially in the case of exoskeleton control, can injure the pilot. Typically, there are three levels of exoskeleton control. The lowest level is a motor driver, to control the speed or torque of the motor. The second level is a robot impedance or admittance control to allow an easy maneuver of the exoskeleton. The impedance control is a robotic pilot model-based control, which would be too difficult if one considers modeling the pilot. Most the available literature has resolved to a reduced model with a single joint with two links [4,5,6]. This inaccurate model is inadequate to handle the situation. Even a slight change in the upper body posture can significantly change the loading condition on the exoskeleton and render the model useless. The wrong interpretation of body inertia can cause the exoskeleton to exert too much torque and cause injury to the pilot. Too little effort, on the other hand, can force the pilot to push the exoskeleton too hard, resulting in excessive torque that causes injury. To accommodate this problem, people have added a third level of control to strategically plan the motion trajectory for various tasks, such as walking, sitting, and standing. Of course, prior knowledge of the driver’s intent is necessary for the controller to issue the appropriate command. Without this knowledge, exoskeletons at present have resolved to have the pilot switch operating modes for changing operating trajectories. This operation is highly inconvenient and is not suitable for regular assistive robot applications. The pilot’s intent detection becomes necessary for the convenient use of the exoskeleton.

The data for analysis in this study collect the body joint-related signals, such as the angles, velocities, torques, and upper body attitude. Body image-based algorithms, such as optical flow and many similar methods [7,8,9], require fixed data collection setups and are unsuitable for mobile applications. The electromyography (EMG)-based method [10] produces very noisy signals and is not ready at this stage. As a result, the inertial measurement units (IMUs) [11]—based system, which is wearable and allows the subject to move around with data gathering, is more suitable for exoskeleton use.

With the collected data, an intuitive approach is translating the captured signal into body postures and motion data and then establishing corresponding thresholds for different movements. These thresholds can be magnitude levels in time, frequency, or the combined time–frequency domain, such as the wavelet analysis. One may notice that these methods are either static, unsuitable for motion identification, or time-sequence analysis-based, requiring seconds to collect enough data for a meaningful analysis. On the other hand, machine learning (ML) methods can match short irregular signal patterns into various tags and are catching much attention for motion recognition research. ML methods can be bases on static data or dynamic time series. K-nearest neighbours, decision tree, and support vector machine (SVM) are based upon static data [12,13,14]. More recently, researchers have proposed the use of deep neural networks, such as convolutional neural networks (CNNs) [15] and recurrent neural networks (RNNs) [14] for recognition. The CNN is still a static-data-based network and the RNN result is still geared toward large motion trajectories. The result by Ragni et al. [16] addressed intent recognition, but was limited to predicting the subject’s choice in three possible ways. Additionally, much of their work was dedicated to deciding if the subject was healthy or a post-stroke patient. The study by Li et al. [17] was about intent prediction, but was about predicting the pitcher’s choice of pitching target. Soliman et al. designed a lower extremity robot for emulating gait maneuvers. They also indicated the possible use of intent control. Additionally, their search was limited to one of the nine targets. Hao et al. [18] used an IMU-based sensory control method for gait synchronization. They used a force sensor to detect robot-limb contact, an IMU sensor to detect toe-off, and an accelerometer to detect heel strike. All the detections were based on signal thresholds. The system by Ji et al. [19] was also event-triggered by a unique intention recognition controller. Their system was a wheeled walker with strings and force sensors attached to the subject. The detection was based on a kinematic force model to determine the direction of the intended motion. Moreover, the system was limited to one-degree-of-freedom yaw motion direction. To the best of the authors’ knowledge, there is, to date, no discussion on how rapidly an algorithm can detect a change in human gait with measured signals.

This study aims to investigate the time required to identify a change in gait or movement and identify the intended gait or posture to enable the system to switch to the corresponding new trajectory rapidly. This rapid automatic recognition of the new gait enables a more comfortable ride without requiring the driver to switch manually among preprogrammed modes. The response time in this research represents the time needed for the system to collect sufficient data for the algorithm to establish recognition. We chose IMU as a wearable sensor due to its non-invasive nature and ease of use with existing exoskeletons. To establish the rapid identification of the change in gait, we introduce the motion transition phase to the detection labels in addition to the commonly used labels of “walking” or “sitting”.

This paper tests two machine learning methods for motion transition detection: the linear feedforward neural network (FNN) and the long short-term memory (LSTM). The subjects wore the IMUs on their waist and right leg and performed three activities: standing, walking, and sitting. The experiments recorded the IMU data and calculated the subjects’ joint angular displacements, angular velocities, and angular accelerations. The experimental results show that 1. The ML networks can rapidly and accurately detect the motion of the subject. 2. With the introduction of the transition phases, the detection time is as fast as 0.17 s when the subject changes from walking to sitting. The detection takes as long as 1.2 s from standing to walking. 3. The results also show that the network trained for one person can apply to different persons without deteriorating the performance. 4. The study also examines the effect of different sampling rates and tests various feature selections for the machine-learning process.

## 2. The Experimental Setup

### 2.1. Participants

A convenience sample of five healthy subjects was recruited from July 2021 to February 2022. None of the subjects had current or previous neurological or orthopedic pathologies of the right leg, and all provided informed consent to participate. Table 1 lists the information on the subjects.

### 2.2. Equipment

In this paper, we used STT-IWS iSen 3.0 as our inertial measurement unit (IMU) system. Each STT-IWS sensor has nine-degrees-of-freedom gyroscopic and magnetometer data. The sensor’s static roll, pitch, and yaw accuracy are all smaller than 2°, with a maximum sampling rate of 400 Hz. One can obtain instance angles of sagittal, transverse, and coronal planes with the subjects wearing the sensors on the right leg. iSen system gathers the wireless IMU signals and pre-processes them into joint angles. The angles are expressed as quaternions and can be transmitted to the computer through the user datagram protocol (UDP). Obtaining the raw IMU data for even faster data collection is also possible; however, the computed angular data rate is rapid enough for this application, and we decided to use the angular data for the analysis.

### 2.3. Experiment Procedure

To begin the data collection, we attached four STT-IWS sensor units to the subject’s right leg, placing them on the sacrum, right thigh, right calf, and right foot. Figure 1 shows how we mounted the sensors on the subject. Then, the STT iSen system was turned on to ensure successful signal capture. At this point, one could set the sampling rate to either 25 or 100 Hz and initiate the data collection. One may notice that the sampling rates are lower than most robotic controllers; however, they are sufficient for human motion.

Because the sensors are hand-tied to the subjects, one must perform a “sensor-to-segment calibration”. The calibration procedure for SST-IWS is very straightforward. Once the locations of the sensors are keyed into the system, the subject only has to stand in a pose that matches the system’s avatar, and the system conducts the calibration on its own.

Once calibrated, the system captures the IMU response signals within the sampling period and performs inverse kinematic calculations to obtain the real-time human posture according to the body dimensions of the pilot. To manifest the detection performance, we designed a sequence of changing movements for the subject to perform to cover the changes in the subjects’ gaits. This research instructed the subject to stand straight to allow the system to recognize his initial state. Then, we asked him to perform a “stand–walk–stand–sit” cycle for five to six minutes. During this cycle, “Stand” represented standing straight as the initial state for 15 s, “Walk” indicated walking around a classroom for 15 to 20 s at his usual pace, and “Sit” involved sitting in a chair for 15 s. Finally, we extracted the data and conducted further analyses.

### 2.4. Data Processing

Data processing was executed using an Intel^®^ CoreTM i7-4790K processor (4.00 GHz) on a machine running a Windows 10 Enterprise operating system. Following the data extraction, we inputted the data into our machine learning system, which was programmed in Python. Once the machine learning calculations were complete and the prediction results were obtained, we transmitted the data to the MATLAB environment for data post-processing and the final analysis, as depicted in Figure 2.

As illustrated in the previous section, we outfitted our subject with four STT-IWS sensor units positioned on the sacrum, right thigh, right calf, and right foot. At this point, one must decide if the detection system uses the sensor signals directly or if we first extract useful features from the signals. Considering the signals from the IMU were in the form of the quaternion, it was hard for machine learning algorithms to interpret the meaning of the quaternion directly. In this research, we decided to pre-process the signals into the motion information of the human body. The STT iSen system computed the subject’s flexion and extension angles in the sagittal plane from the captured signals [20]. One could also obtain the angles of the pelvis, right hip, right knee, and right ankle in the sagittal plane. In addition, the system also provided the first and second derivatives of these angles to obtain the angular velocity and angular acceleration of the subjects’ joints in their sagittal planes. This research adopted these parameters as the features to train the machine learning system.

The following sections describe the Python-based ML algorithm developed in this research.

### 2.5. Intent Detection

This research developed two programs for two different algorithms, a linear feedforward neural network (FNN) and a long short-term memory (LSTM), to predict the switching among three common human actions: standing, walking, and sitting. The ML training programs were developed in the Python environment.

In the previous session, the signals of the subject measured by the IMU had three kinds of features, the angles, angular velocity, and angular acceleration of the subject in his sagittal plane, and each signal set included the data of the pelvis, right hip, right knee, and right ankle of the subject. These 12 features were treated as the inputs of our ML training system, and the outputs were the five labels of human actions: standing, walking, sitting down, sitting, and standing up. Notice that instead of the three actions, we introduced two additional labels for “sitting down” and “standing up” for reasons that are explained later.

Then, we chose either FNN or LSTM as our learning algorithms. The FNN used three fully connected layers to run the ML system (Figure 3). There were 12 input nodes in the first layer because it contained the data of three kinds of features in the four joints of the subject. The remaining inputs and outputs of other layers were set as five because we needed the output to match the five labels of human actions. Before feeding the input to the fully connected layers, we inserted a dropout layer with p=0.2, where p was the probability of the layer forcing the input to be zeroed. The purpose of the dropout layer was to reduce the chance of overfitting. After being operated by the fully connected layers, the output would run through a softmax layer to ensure that all outputs lay within [0, 1] and its summation equaled 1 to match the probability distribution. The output could be easier to discriminate after passing through the softmax layer.

The LSTM used a 2-layer LSTM as the main structure (Figure 4). The reason for a 2-layer LSTM was the computing limitation. A 2-layer LSTM here could cause the entire network to be too complicated to calculate. The input of the LSTM layer was 12, and the remaining inputs and outputs of other layers were 5 for a reason similar to the FNN. In addition to dropout layers and a softmax layer, we added a batch normalization layer and a ReLU layer to the LSTM system. The batch normalization layer helped avoid overfitting similar to dropout layers, while the ReLU layer helped eliminate negative terms. Some important parameters of the FNN and LSTM ML systems are shown in Table 2. The study aimed to detect the change in the pilot’s motion rapidly. The data size, which affected the time required to gather enough data, was limited. We used the same batch size and number of epochs for a fair comparison. We also used the same optimizer to check the learning performance. Because of the limited number of volunteers, we carefully separated the data for training and validation. The learning rate and weight decay are the results of a long process of iterations.

## 3. Results

### 3.1. Data and Labels

Table 3 provides comprehensive information regarding the training and testing data. To begin with, we recorded a total time length of 3630 s for the training data of subject 1. Depending on different tests, the sampling rate was set to either 25 or 100 Hz. The interval for each human activity, namely, standing, walking, and sitting, was approximately 15 s. Accordingly, the “stand–walk–stand–sit” cycle took approximately one minute. Subsequently, we obtained testing data from various individuals with different sampling rates, as outlined in Table 3. We asked the subjects to perform the “stand–walk–stand–sit” cycle for five to six minutes during testing. It is worth noting that subject 1, who was also included in Test1, Test4, and Test7, was the same individual as the one in the training data.

Labeling the data before inputting the training and testing data into the ML networks was necessary. As discussed before, we asked the subjects to conduct the sequence of motion at specific time stamps and hand-labeled the signal. To accurately recognize the transition between movements (standing, walking, and sitting), as mentioned in the previous section, this research developed five labels for human actions: standing, walking, sitting down, sitting, and standing up. We split the “sitting” motion into “sitting down”, “sitting”, and “standing up”. During our research, we noticed that the algorithm could separate “sitting” and “standing” into static postures. Without the transition period, the algorithm tended to emphasize the accuracy of the posture identification, instead of how fast it could detect a posture change. To emphasize the transition between movements, we introduced the two addition labels for the transition phase to force the algorithms to emphasize detecting the transition. Figure 5 shows the variation in one subject’s flexion and extension angles during the test. One can see that all the joint angles share similar time variation characteristics for different movements. There are also significant differences among the actions, particularly for the three movements related to sitting. Figure 5 also shows how we labeled the actions according to the time stamps. Specifically, “sitting down” (abbreviated as “sd” in Figure 5) referred to when the subject shifted their posture from standing to sitting, while “standing up” (“su” in Figure 5) represented the subject’s transition from sitting to standing. Both actions took approximately one second to complete and exhibited distinct performance differences compared to “sitting”. Therefore, it was evident that separating “sitting” into “sitting down, sitting, and standing up” helped differentiate the various stages of movement and warranted the expansion of the three distinct movements into five labels.

Figure 6 shows the identification results. One can see that the network can successfully identify the different actions of the human pilot. The time difference between “stand” and “walk” shows the limitation of around 1 s for the network to detect a change in motion from standing to walking. The network can detect the change from “stand” to “sit” much faster, possibly due to the greater movements involved.

### 3.2. FNN versus LSTM

This research first compared the two algorithms, FNN and LSTM, with 25 Hz data. Table 4 presents the information and outcomes obtained using both algorithms. In Table 4, $T_{stand}$ indicates the point in time when the subject commenced or ceased standing, while ${∆T}_{stand}$ represents the time difference between the ML system’s calculated $T_{stand}$ and the actual $T_{stand}$. Similarly, ${∆T}_{walk}$ and ${∆T}_{sit}$ indicate the time difference between the moment the subject began or concluded walking/sitting.

Based on the data presented in Table 4, we can observe that both methods achieve favorable outcomes, with an accuracy of above 80%. Additionally, LSTM was notably more accurate than FNN across all ∆T and accuracy measures, though LSTM also required a lengthier training time for all subjects. Furthermore, the operating time for LSTM in the MATLAB environment was slightly more prolonged than that for the FNN. The reason behind LSTM’s superior accuracy and longer training time is that it is a more complex ML algorithm that accounts for the time as another parameter. As the objective of this paper was to detect human intent and compute the time when the subject altered their action, LSTM, a type of recurrent neural network (RNN), was more appropriate than the FNN in this scenario.

The confusion matrix subject 1 in Table 5 shows that the detection for the posture reaches 94% for “stand” and “sit” and 89% for “walk. The most confusion came from “walk” to “stand” with an 11% error, while the lowest error rate of 0.5% was achieved for “sit” to “walk”.

### 3.3. Different Sampling Rates

Subsequently, we conducted a comparison between two different sampling rates, 25 and 100 Hz, while utilizing the LSTM algorithm. Table 6 displays the information and results of the two different sampling rates. As shown in Table 5, it is evident that a higher sampling rate leads to greater accuracy and lower ∆T across all subjects. However, it came at the cost of a longer training time, and the required training time was proportionate to the ratio of the sampling rate in both cases. Notice that the LSTM network in this application is for processing time series. A closer examination of the series shows that human intention can be very random. There is no rule against the pilot taking any particular action other than that there must be a transition between actions. In this case, the advantage of LSTM over FNN becomes less significant. As the network receives the motion features from the subject, it analyzes the data and judges the current motion. A faster sampling rate provides more detailed information for the motion features and supposedly faster recognition. However, there is a limit to how fast a human can move. The network has to receive enough data samples to detect specific characteristics for making judgments. Faster sampling may pile up the queue with data samples that carry very little information for this purpose, thus providing no use for intention recognition. The 25 Hz sampling is consistent with human vision. That might be the reason iSen chose this speed. From the vision point of view, higher sampling rates may not be helpful. The 100 Hz sampling may provide redundant information; although, it enables the system to detect the change at a faster rate. Additionally, the operating time in MATLAB was slightly longer in the higher sampling rate case. Overall, we observed that the accuracy increased from 88% to 95% when we increased the sampling rate from 25 to 100 Hz.

### 3.4. Different Subjects

Finally, we compared five subjects using the LSTM algorithm, and the sampling rate was 100 Hz. Table 6 displays the information and results obtained for the five different subjects, with subject 1 being the same person as the trainer. As shown in Table 7, it is evident that good results are achieved across all subjects, with accuracies surpassing 92%. The results of this scenario demonstrate that our ML system can rapidly and accurately detect when there is an intention to switch motion, even when the training and testing data are obtained from different individuals with varying genders, ages, and physical conditions. We acknowledge that the practice is irregular to use the network trained by one set of data to apply to a different application. However, for the exoskeleton application, it is difficult to make the pilot wear an untrained suit without some prior capability to adapt to the person. The ability of the network to perform based on the data obtained from one subject enables the system’s initial setup. Then, it can take time to learn the characteristics of various pilots.

## 4. Conclusions

This paper described the successful development of a machine learning system for human intent detection (HID) using two algorithms: linear feedforward neural network (FNN) and long short-term memory (LSTM). The system detected transitions between three common human movements (standing, walking, and sitting) using data from four inertial measurement units (IMUs) attached to the subject’s right leg. This research also proposed to distinctively separate the transition before and after sitting into “sitting down” and “standing up” to highlight the ability to detect a change in the pilots’ intent. The results show that both algorithms achieve good accuracy, with LSTM outperforming the FNN in terms of the average time difference and accuracy; although, it takes longer to train. The identification accuracy of the two structures was above 80%, while LSTM could improve the accuracy from 82.5% to 88%. The accuracy is better than the previous results of 83.3% using a single inertial sensor [21]. The average time difference and accuracy of LSTM were better than the FNN; however, it also took longer to train. Because training time is irrelevant to the application run time, LSTM is more suitable than FNN.

This paper showed that ML networks could: 1. Rapidly and accurately identify the motion of the subject. 2. The introduction of the transition phase helped rapidly detect the change in motion with a detection time as fast as 0.17 s when the subject changed from walking to sitting. The detection could be as long as 1.2 s for the transition between standing and walking. 3. The network trained for one person could apply to different persons without considerable changes in performance. 4. The study also compared different sampling rates and found that higher rates led to improved accuracy and a lower detection time but with a longer training time.

The IMU system is still limited by causality; therefore, one can only try to detect the changes as soon as possible after the pilot starts the movement. It would be desirable to be able to foretell the pilot’s intention. Future research will measure the signals from the core muscles to investigate the possibility of detecting core muscle signals and using them for advanced detection purposes.

## Figures and Tables

**Figure 1 sensors-23-07203-f001:**
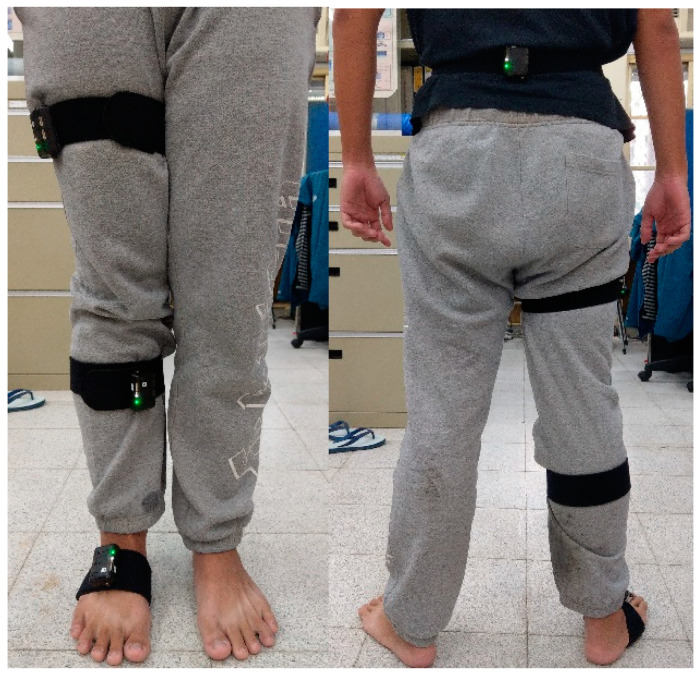
The front and back views of the subject wearing STT-IWS sensors on the sacrum, right thigh, right calf, and right foot of the right leg.

**Figure 2 sensors-23-07203-f002:**
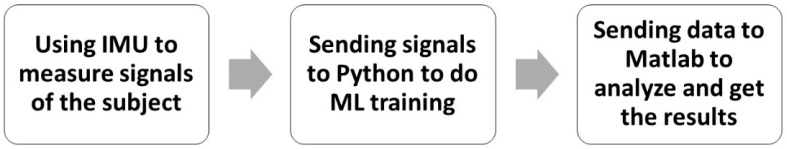
The schematic diagram.

**Figure 3 sensors-23-07203-f003:**
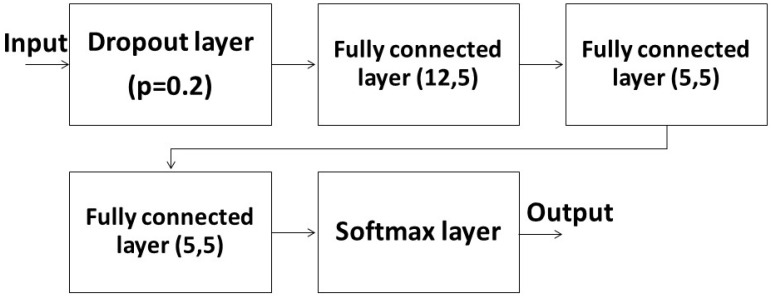
The structure of FNN ML system.

**Figure 4 sensors-23-07203-f004:**
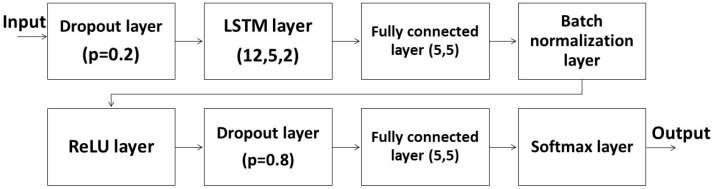
The structure of the LSTM ML system.

**Figure 5 sensors-23-07203-f005:**
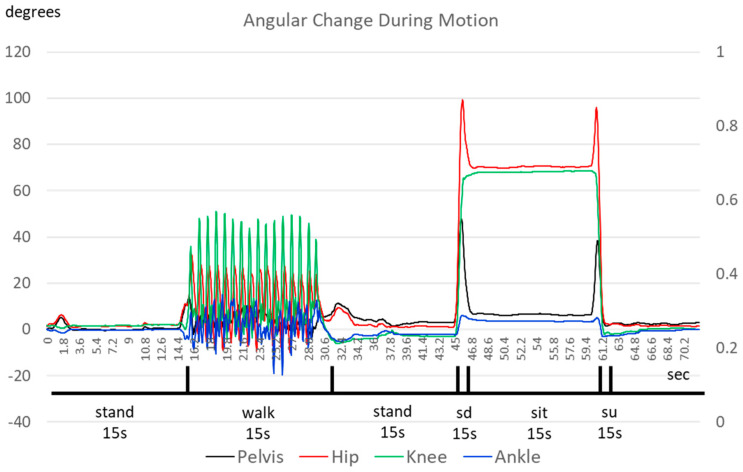
The angles in the sagittal plane between different human actions.

**Figure 6 sensors-23-07203-f006:**
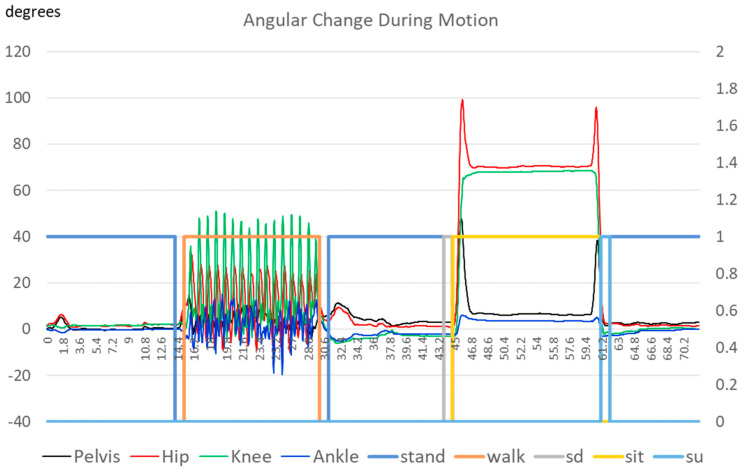
The results of the angles in the sagittal plane between different human actions.

**Table 1 sensors-23-07203-t001:** The information of the five healthy subjects.

Subject	1	2	3	4	5
Sex	male	male	male	male	female
Height (cm)	173	171	174	178	168
Weight (kg)	70	68	80	80	62
Age	29	23	23	33	23
Length of right thigh (cm)	47	44	43	49	46
Length of right calf (cm)	40	38	43	49	42
Length of right foot (cm)	26	25	25	29	25

**Table 2 sensors-23-07203-t002:** The parameters of the ML system.

ML System	FNN	LSTM
Training/validation ratio	2:1	2:1
Number of epochs	150	150
Batch size	10	10
Optimizer	AdamW	AdamW
Learning rate	0.001	0.001
Weight decay	0.00001	0.00001
Loss function	Mean square error loss	Cross entropy loss

**Table 3 sensors-23-07203-t003:** The information for our training and testing data.

	Subject	Algorithm	Sampling Rate (Hz)	Period (s)	Training Time in Python (h)	Operating Time in MATLAB (s)
Training	1		25 or 100	3630		
Test1	1	FNN	25	375	0.65	3
Test2	2	FNN	25	315	0.95	3
Test3	3	FNN	25	315	0.95	3
Test4	1	LSTM	25	375	1.37	5
Test5	2	LSTM	25	315	1.4	5
Test6	3	LSTM	25	315	2.05	7.5
Test7	1	LSTM	100	315	5.36	8.5
Test8	2	LSTM	100	315	5.4	8.75
Test9	3	LSTM	100	315	5.38	8.8
Test10	4	LSTM	100	390	5.98	8.5
Test11	5	LSTM	100	375	5.92	8.5

**Table 4 sensors-23-07203-t004:** The information of results between two different algorithms.

	Test1	Test2	Test3	Test4	Test5	Test6
Subject	1	2	3	1	4	5
Algorithm	FNN	FNN	FNN	LSTM	LSTM	LSTM
Sampling rate (Hz)	25	25	25	25	25	25
Period (s)	375	315	315	375	315	315
Training time in Python (h)	0.65	0.95	0.95	1.37	1.4	2.05
Operating time in MATLAB (s)	3	3	3	5	5	7.5
Average ${∆T}_{stand}$ (s)	1.324	1.314	1.355	0.86	0.8535	0.88
Average ${∆T}_{walk}$ (s)	1.865	1.813	2.05	1.21	1.171	1.488
Average ${∆T}_{sit}$ (s)	0.863	0.678	1.101	0.56	0.436	0.72
Accuracy (%)	82.80	83.75	81.39	88.83	89.48	87.40

**Table 5 sensors-23-07203-t005:** The confusion matrix for subject 1.

	Actual	Stand	Walk	Sit
Predict				
stand		0.9484	0.0516	0
walk		0.1099	0.8901	0
sit		0.0564	0.0048	0.9388

**Table 6 sensors-23-07203-t006:** The information of results between two different sampling rates.

	Test4	Test5	Test6	Test7	Test8	Test9
Subject	1	2	3	1	4	5
Algorithm	LSTM	LSTM	LSTM	LSTM	LSTM	LSTM
Sampling rate (Hz)	25	25	25	100	100	100
Period (s)	375	315	315	375	315	315
Training time in Python (h)	1.37	1.4	2.05	5.36	5.4	5.38
Operating time in MATLAB (s)	5	5	7.5	8.5	8.75	8.8
Average ${∆T}_{stand}$ (s)	0.86	0.8535	0.88	0.3825	0.36	0.3265
Average ${∆T}_{walk}$ (s)	1.21	1.171	1.488	0.62	0.444	0.858
Average ${∆T}_{sit}$ (s)	0.56	0.436	0.72	0.281	0.355	0.168
Accuracy (%)	88.83	89.48	87.40	94.67	95.18	94.67

**Table 7 sensors-23-07203-t007:** The information of results between five different subjects.

	Test7	Test8	Test9	Test10	Test11
Subject	1	2	3	4	5
Algorithm	LSTM	LSTM	LSTM	LSTM	LSTM
Sampling rate (Hz)	100	100	100	100	100
Period (s)	375	315	315	390	375
Training time in Python (h)	5.36	5.4	5.38	5.98	5.92
Operating time in MATLAB (s)	8.5	8.75	8.8	8.5	8.5
Average ${∆T}_{stand}$ (s)	0.3825	0.36	0.3265	0.3371	0.4542
Average ${∆T}_{walk}$ (s)	0.62	0.444	0.858	1.2458	0.8817
Average ${∆T}_{sit}$ (s)	0.281	0.355	0.168	0.3175	0.2642
Accuracy (%)	94.67	95.18	94.67	92.83	93.43

## Data Availability

The test data will be made available upon request.

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
