# Peer review of "Machine Learning for Human Motion Intention Detection"

_sensors, 2023, doi:10.3390/s23167203_

Round 1

Reviewer 1 Report

Paper's novel contributions may be provided in a point-wise manner.

Are the collected data put in public repository for other researchers to reproduce the experiment? If yes please mention the link. 

What type of data is collected or how they are processed, the authors should mention it.

Why the authors have considered a particular parameter in Table 2 need to justify it.

Why there is a time variation of the same subject using the same algorithm in Table 3? Justify.

How is the sampling rate impacting the accuracy? Further analysis may be added.

What are the limitations?

Language may be simplified for better readability.

Author Response

Comments and Suggestions for Authors

Paper’s novel contributions may be provided in a point-wise manner.

We have rewritten the abstract and most of the introduction, and the conclusion (marked in red) p.1-3, and p.10. And have listed our contributions in a point-wise manner.

Are the collected data put in public repository for other researchers to reproduce the experiment? If yes please mention the link.

The data download link is

https://drive.google.com/drive/folders/1ZbZGFScVn26bY0djE1O-2lcDpZPYojJ5?usp=sharing

What type of data is collected or how they are processed, the authors should mention it.

The motion data was collected by STT-IWS iSen 3.0. iSen system gathers the wireless IMU signals and pre-processes them into joint angles. The angles are expressed as quaternions and can be transmitted to the computer through the User Datagram Protocol (UDP). Obtaining the raw IMU data for even faster data collection is also possible, but the computed angular data rate is fast enough for this application, and we have decided to use the angular data for analysis. The data will be made available upon request.

We have added this information in section 2.2 (p. 3, in red).

Why the authors have considered a particular parameter in Table 2 need to justify it.

The study aims to detect the change in the pilot’s motion quickly. The size of the input, which affects the time required to gather enough data, is limited. We have used the same batch size and number of epochs for a fair comparison. We have also used the same optimizer to check the learning performance. Because of the limited number of volunteers, we have carefully separated the data for training and validation. The learning rate and the weight decay are the results of a long process of iterations. (p.6 paragraph 1)

Why there is a time variation of the same subject using the same algorithm in Table 3? Justify.

The time variation arises from different tests. Although the data are collected from the same subject, the different tests may have varied signal contents and time durations. As a result, there are also differences in the training time required.

How is the sampling rate impacting the accuracy? Further analysis may be added.

Indeed the faster sampling time has an influence but in a contradicting way. The LSTM network in this application is for processing time series. The human intention is usually very random. There is no rule against the pilot taking any particular action other than that there must be a transition between actions. The advantage of LSTM over FNN is thus not as significant. As the network receives the motion features from the subject, it analyzes the data and judges the current motion. A faster sampling rate provides more detailed information on the motion features and supposedly faster recognition. However, there is a limit to how fast a human can move. The network has to receive enough data samples to detect specific characteristics for making judgments. Faster sampling may pile up the queue with data samples that carry very little information for this purpose, thus providing no use to intention recognition. The 25Hz sampling is consistent with human vision. That might be the reason iSen chose this speed. From the vision point of view, higher sampling rates may not be helpful. The 100 Hz sampling may provide redundant information, although it enables the system to detect the change faster.

We have rewritten section 3.3 to explain the situation better.

What are the limitations?

The IMU sensors only pick up signals after they have happened. In other words, the system is still limited by causality. We can only try to detect the changes as soon as possible after the pilot has started the movement. It would be desirable to actually be able to foretell the pilot’s intention. The physiatrist on our team told us that the core muscles act before the motion takes place. We are now looking into the possibility of detecting the signals and using them for the detection. We have added more discussion in the discussion section. (P. 9, last paragraph)

Comments on the Quality of English Language

Language may be simplified for better readability.

We have rewritten the abstract, most of the introduction, and a significant portion of the introduction to the algorithm. We believe the revised manuscript should be much easy to follow. (P. 1, 2, 3, end of 5-6, p. 8, p. 9 last paragraph)

Reviewer 2 Report

In this note, the authors identify the transitions between different human actions, such as standing, walking, and sitting. They construct two machine learning networks, a linear Feedforward Neural Network (FNN) and a Long Short-Term Memory (LSTM), using data from inertial measurement units (IMU) worn on the subjects' right leg. The results show that FNN and LSTM perform well in predicting the transitions, with LSTM achieving minor errors but requiring longer training time. The work has significant contributions with well-presented experimental setups. However, there are a few minor concerns that need to be addressed before publication:

1. The placement and calibration of IMU sensors should be presented.

2. The authors are suggested to show the pseudo code/algorithms for FNN and LSTM inspired by the Python environment. This will improve the better readability of the note.

3. The authors have presented different actions in Fig. 7, separated by black lines. It is unclear whether the authors recorded pair of activities in continuations or at different intervals and added them afterward. This affects the outcomes significantly.

4. There should be some graphical representation of results in addition to the tables.

Author Response

The work has significant contributions with well-presented experimental setups. However, there are a few minor concerns that need to be addressed before publication:

Thank you for the comment. We have addressed all the comments, and with your help, the manuscript is now in a much better form for presentation.

1. The placement and calibration of IMU sensors should be presented.

Yes, sensor calibration is essential to obtain meaningful data from the subjects. iSen provides the user with a “sensor-to-segment calibration” procedure to follow. Once the locations of the sensors are keyed into the system, the subject only has to stand in a pose that matches the system’s avatar, and the system conducts the calibration on its own. (p.4, 2nd paragraph).

2. The authors are suggested to show the pseudo code/algorithms for FNN and LSTM inspired by the Python environment. This will improve the better readability of the note.

Thank you for the comment. We have addressed all the comments, and we believe, with your help, the manuscript is now in a much better form for presentation.

3. The authors have presented different actions in Fig. 7, separated by black lines. It is unclear whether the authors recorded pair of activities in continuations or at different intervals and added them afterward. This affects the outcomes significantly.

Yes, it is crucial that the test process reflects the actual motion. The test procedure for the subjects is a series of continuous movements to reflect the actual situation. In other words, the sequence from standing still, walking, sitting, and standing up is a continuous sequence to test the ability of the network to separate different movements in practical life. However, this design does pose a future question of how we define the period of various movements. In this study, we asked the subjects to conduct the motion change at some specific time stamps and hand-labeled the signal. We are still investigating a better way to label the signals. 

4. There should be some graphical representation of results in addition to the tables.

Thank you for the comment. We have addressed all the comments, and we believe, with your help, the manuscript is now in a much better form for presentation.

Reviewer 3 Report

Overall, the authors examine whether change detection between activities (sitting, walking, standing) can be recognized using data from IMUs, with a motivation for use in exoskeleton. They achieve reasonanble success, the paper requires providing more details about how the classification was done, better presentation of the results (e.g. include a confusion matrix), and requires more comparisons to other studies.

The authors should also make all the data sets and analysis code freely available (and not "available upon request") unless there is a compelling reason not to, in the name of reproducability

Specific comments:

The affiliations are unclear - no need to write position (PhD student, professor, etc.) and the affiliations should be in numerical order

Abstract - as the motivation for the study is the use of exoskeletons, this should be included in the abstract

- line 20 - the IMUs do not measure joint angle, velocity, etc. rather they are calculated based on data from >1 IMU

Line 41 - Commonly, a slight movement in the pilot's joint often involves huge torque forces from the joints.

* it is not clear to me what you mean by pilot

* it is not obvious to me why this sentence is true

line 73 - these are not "commercial systems" but types of algorithms

lines 89-91 - these claims are not convincingly made. Citations would be helpful

line 101 - the review of previous studies should include studies specifically looking at change detection (the goal of this study)

line 118 - todate - should be two words (to date)

line 128-129 "very few papers have examined their performance as wearable gait sensors" - this statement is not true, a quick search on google scholar will reveal a large number of papers using IMUs and other wearable sensors for studying different aspects of gait

line 132-133 "no application of IMU for HID based on machine learning" - this claim is also not true - a quick literature search reveals many recent papers that do this

Table 1 - remove last names from the table - this do not contribute to the analysis and may reveal who the participants are

Gender - you probably mean sex and not gender

line 167 - what is the "human gesture"? maybe you mean posture? be specific as to what it is calculating, and how it is doing this (Kalman filter?)

Figure 2 - add to the caption details of where the sensors are placed

Figure 3 is unneccesary (Figure 2 is sufficient)

line 192 - it is unclear what about IMU signals is "primitive" - how do you know this is the case? did you test it?

Table 3 - again, remove subject names

line 253 - "label the data" - how was this done?

Figure 7 - what are the units? What are the different colors? 

Table 4 - again, remove subject names

page 10 - it would helpful for the best results to also include a confusion matrix

line 331 - "irelativent" - I think you mean irrelevant

Conclusions - it is not clear how long the identification takes (~8s - is this the Matlab operating time?) Is this system suitable for use in real-time applications such as an exoskeleton?

- the conclusions / discussion should compare the accuracy rates to other studies of intent detection

- Why were 5 subjects selected?

- The references are included twice

- references 4,5,7,11 - providing just a DOI will be more useful than also the long URL

- reference 22 is missing the date

The english is generally fine, there are some spelling mistakes and use of wrong words. Editing using software such as grammarly is recommended

Author Response

Comments and Suggestions for Authors

Overall, the authors examine whether change detection between activities (sitting, walking, standing) can be recognized using data from IMUs, with a motivation for use in exoskeleton. They achieve reasonanble success, the paper requires providing more details about how the classification was done, better presentation of the results (e.g. include a confusion matrix), and requires more comparisons to other studies.

The authors should also make all the data sets and analysis code freely available (and not “available upon request”) unless there is a compelling reason not to, in the name of reproducability

Author response:

Thank you for the helpful comments. We have rewritten most of the manuscript, added the confusion matrix (Table 5), and added new discussions on the more recent research.

Also, we will provide the download at

https://drive.google.com/drive/folders/1ZbZGFScVn26bY0djE1O-2lcDpZPYojJ5?usp=sharing

Specific comments:

The affiliations are unclear - no need to write position (PhD student, professor, etc.) and the affiliations should be in numerical order

Corrected as advised.

Abstract - as the motivation for the study is the use of exoskeletons, this should be included in the abstract

Thank you for the helpful comments. We have rewritten the abstract and explained the algorithm’s use with the exoskeleton within. The revised abstract-

The gait pattern of the exoskeleton control conflicting with the human operator’s (the pilot) intention may cause awkward maneuvering or even injury. Therefore, it has been the focus of many studies to help decide the proper gait operation. However, the timing for the recognization plays a crucial role in the operation. Delayed detection of the pilot’s intent can be equally undesirable to the exoskeleton operation. Instead of recognizing the motion, this study examines the possibility of identifying the transition between gaits to achieve in-time detection. This study used the data from IMU sensors for future mobile applications. Further, we tested using two machine-learning networks: a linear Feedforward Neural Network and a long short-term memory network. The gait data are from five subjects for training and testing. The study results show that: 1. The network can successfully separate the transition period from the motion periods. 2. The detection of gait change from walking to sitting can be as fast as 0.17 seconds which is adequate for future control applications. However, detecting the transition from standing to walking can take as long as 1.2 seconds. 3. This study also found that the network trained for one person can also detect movement changes for different persons without deteriorating the performance. (abstract)

- line 20 - the IMUs do not measure joint angle, velocity, etc. rather they are calculated based on data from >1 IMU

Indeed, we have corrected the narration as follows:
Further, we tested using two machine-learning networks: a linear Feedforward Neural Network and a long short-term memory network. The gait data are from five subjects for training and testing. The study results show that: 1. The network can successfully separate the transition period from the motion periods. 2. The detection of gait change from walking to sitting can be as fast as 0.17 seconds which is adequate for future control applications. However, detecting the transition from standing to walking can take as long as 1.2 seconds. 3. This study also found that the network trained for one person can also detect movement changes for different persons without deteriorating the performance. (abstract)

Line 41 - Commonly, a slight movement in the pilot’s joint often involves huge torque forces from the joints.

* it is not clear to me what you mean by pilot

The exoskeleton operator is often called the “pilot,” as termed in many references [1]. We have added the explanation (first line in the abstract.)

* it is not obvious to me why this sentence is true

I was trying to explain the hierarchy of control. I agree that the sentence by itself isn’t convincing. I have replaced it with a more detailed explanation of the control hierarchy of the exoskeleton as follows:
Typically, there are three levels of exoskeleton control. The lowest level is a motor driver, to control the speed or torque of the motor. The second level is a robot impedance or admittance control to allow an easy maneuver of the exoskeleton. The impedance control is a robotic pilot model-based control, which would be too difficult if one considers modeling the pilot. Most available literature has resolved to a reduced model with a single joint with two links [4-6]. This inaccurate model is inadequate to handle the situation. Even a slight change in the upper body posture can significantly change the loading condition on the exoskeleton and render the model useless. The wrong interpretation of the body inertia can cause the exoskeleton to exert too much torque and cause injury to the pilot. Too little effort, on the other hand, can force the pilot to push the exoskeleton too hard, resulting in excessive torque that causes injury. To accommodate this problem, people have added a third level of control to strategically plan the motion trajectory for various tasks such as walking, sitting, and standing. (p.1, lines 34-47)

line 73 - these are not “commercial systems” but types of algorithms

Our apology; we were mixing algorithms with systems. We have corrected the description in the revision (p.2, line 57)

lines 89-91 - these claims are not convincingly made. Citations would be helpful

I agree. The sentence is not convincing by itself. We have added new citations and used numerical numbers to explain. (p.3, lines 66-68)

line 101 - the review of previous studies should include studies specifically looking at change detection (the goal of this study)

We have added new references that specifically address the changes in human intention and have discussed their different intentions. (Ref [18][19][20] in the manuscript)

line 118 - todate - should be two words (to date)

Corrected as advised.

line 128-129 “very few papers have examined their performance as wearable gait sensors” - this statement is not true, a quick search on google scholar will reveal a large number of papers using IMUs and other wearable sensors for studying different aspects of gait

We agree. We were not careful enough in making that statement and have removed the comment.

line 132-133 “no application of IMU for HID based on machine learning” - this claim is also not true - a quick literature search reveals many recent papers that do this

We apologize. We have removed the statement and have used a more specific description for our contribution. 

Table 1 - remove last names from the table - this do not contribute to the analysis and may reveal who the participants are

Indeed, thank you for the advice. We have removed the personal identifications in the tables.

Gender - you probably mean sex and not gender

Yes, we have changed it as advised.

line 167 - what is the “human gesture”? maybe you mean posture? be specific as to what it is calculating, and how it is doing this (Kalman filter?)

  1. Thank you for pointing that out. We have corrected the words (and after that). (line 144)
  2. The system does an inverse kinematic calculation for the postures. We have rewritten the paragraph to explain the details. (lines 143-149)

Figure 2 - add to the caption details of where the sensors are placed

We have added the description.

Figure 3 is unneccesary (Figure 2 is sufficient)

We have removed the figure.

line 192 - it is unclear what about IMU signals is “primitive” - how do you know this is the case? did you test it?

I agree that this can be confusing. The IMU signals are in the quaternion form. It is not as easily interpreted directly. We have rephrased the sentence. (lines 168-170)

Table 3 - again, remove subject names

Corrected.

line 253 - “label the data” - how was this done?

To reflect the actual condition in the application, we have designed the test procedure for the subjects in a series of continuous movements. This design then induced the problem of how we define the period of various movements. In this study, we asked the subjects to conduct the motion change at some specific time stamps and hand-labeled the signal. We are still investigating a better way to label the signals. We have rewritten the paragraph to explain the process in detail. (lines 235-237)

Figure 7 - what are the units? What are the different colors?

Sorry for the miss. We have added the labels with units.

Table 4 - again, remove subject names

Corrected

page 10 - it would helpful for the best results to also include a confusion matrix

We have added the confusion matrix of our result for subject 1 to show that the network achieves 95% detection accuracy for “stand,” 89% for “walk,” and 94% for “sit” (Table 5).

line 331 - “irelativent” - I think you mean irrelevant

Corrected! Sorry for the typo.

Conclusions - it is not clear how long the identification takes (~8s - is this the Matlab operating time?) Is this system suitable for use in real-time applications such as an exoskeleton?

The identification is in real-time, so the calculation time is well under 10 ms. The Matlab time we mentioned in the previous version is the training time.

- the conclusions / discussion should compare the accuracy rates to other studies of intent detection

Author response:

We compared the result with a 2023 result in [23] of the manuscript. The authors used only a single inertial sensor but did not mention how fast their network could detect the change. (lines 338-343)

- Why were 5 subjects selected?

At this point, the subjects are all from our laboratory and have all signed the research agreement form approved by the IRA review board. We will be collecting more data from more subjects in the future.

- The references are included twice

Our apology. Corrected.

- references 4,5,7,11 - providing just a DOI will be more useful than also the long URL

We have corrected the format accordingly.

- reference 22 is missing the date

Corrected.

Comments on the Quality of English Language

The english is generally fine, there are some spelling mistakes and use of wrong words. Editing using software such as grammarly is recommended

We have carefully gone through the revised manuscript with grammarly to make sure it contains no spelling mistakes.

[1]         J. Ghan, R. Steger, and H. Kazerooni, “Control and system identification for the Berkeley lower extremity exoskeleton (BLEEX),” Advanced Robotics, vol. 20, no. 9, pp. 989-1014, 2006/01/01 2006, doi: 10.1163/156855306778394012.

Round 2

Reviewer 1 Report

Please check the alignment of the tables and figures.

The authors have responded to all my queries. After minor editing the paper may be accepted.

Reviewer 3 Report

I am satisfied with the corrections made by the authors and recommend accepting the paper